# The Metabolic Concept of Meal Sequence vs. Satiety: Glycemic and Oxidative Responses with Reference to Inflammation Risk, Protective Principles and Mediterranean Diet

**DOI:** 10.3390/nu11102373

**Published:** 2019-10-05

**Authors:** Niva Shapira

**Affiliations:** Department of Nutrition, School of Health Professions, Ashkelon Academic College, Ashkelon 78211, Israel; nivash@edu.aac.ac.il; Tel.: +972-(0)3-649-7998

**Keywords:** satiety, glycaemia, oxidative stress, postprandial, Mediterranean diet, meal sequence

## Abstract

With increasing exposure to eating opportunities and postprandial conditions becoming dominant states, acute effects of meals are garnering interest. In this narrative review, meal components, combinations and course sequence were questioned vis-à-vis resultant postprandial responses, including satiety, glycemic, oxidative and inflammatory risks/outcomes vs. protective principles, with reference to the Mediterranean diet. Representative scientific literature was reviewed and explained, and corresponding recommendations discussed and illustrated. Starting meals with foods, courses and/or preloads high in innate/added/incorporated water and/or fibre, followed by protein-based courses, delaying carbohydrates and fatty foods and minimizing highly-processed/sweetened hedonic foods, would increase satiety-per-calorie vs. obesogenic passive overconsumption. Similarly, starting with high-water/fibre dishes, followed by high-protein foods, oils/fats, and delayed/reduced slowly-digested whole/complex carbohydrate sources, optionally closing with simpler carbohydrates/sugars, would reduce glycaemic response. Likewise, starting with foods high in innate/added/incorporated water/fibre/antioxidants, high monounsaturated fatty acid foods/oils, light proteins and whole/complex carbohydrate foods, with foods/oils low in n-6 polyunsaturated fatty acids (PUFA) and n-6:n-3 PUFA ratios, and minimal-to-no red meat and highly/ultra-processed foods/lipids, would reduce oxidative/inflammatory response. Pyramids illustrating representative meal sequences, from most-to-least protective foods, visually communicate similarities between axes, suggesting potential unification for optimal meal sequence, consistent with anti-inflammatory nutrition and Mediterranean diet/meal principles, warranting application and outcome evaluation.

## 1. Introduction

As people in the modern world are increasingly in a postprandial state [1], acute effects of meals are garnering expanding interest and concern beyond the morning fast state. Postprandial studies show that common meals may have significant negative effects, including promotion of chronic inflammation, especially in people at high risk [2]. These effects would occur several times daily and last for several hours [3], thus covering a major part of the day, potentially up to 18 h [1].

After food intake, inflammatory markers in the systemic circulation may transiently increase as a reaction to the acute ingestion of nutrients and immune challenges. This postprandial inflammation, which is a normal metabolic response, has become an important risk factor for human health, since people in industrialised countries are almost constantly in a postprandial state, with corresponding inflammation trajectory [4,5].

Chronic low-grade inflammation is a major contributor to non-communicable disorders, including metabolic syndrome, type 2 diabetes and cardiovascular disease [6,7,8], often induced [9,10,11,12,13,14] by prolonged activation of the immune system by overnutrition [15]. 

Metabolic disorders such as obesity, characterised by a chronic low-grade inflammation in the fasting state, increase the magnitude of postprandial inflammation [16]. The degree of postprandial inflammation induced by a meal was shown to depend on its energy value, the amount of carbohydrates, glycaemic index (GI) and glycaemic load (GL) and the amount and composition of lipids [5].

Though meal planning has demonstrated healthier outcomes [17], e.g., as related to control of energy intake [18], and increased food variety and diet quality have been associated with lower obesity and related risk factors [19], its potential for reducing primary health risks has received relatively limited attention in the health care literature [20].

In studies on energy intake and obesity, high-carbohydrate and -fat meals have demonstrated lower satiating effects, leading to overeating, as compared to meals high in innate/added/incorporated water, fibre and protein [21], whereas increased postprandial satiety-per-kilocalorie (kcal) through pre-meal intake of water [22], soups and low-caloric preloads [23,24] reduced intake [25]. Similarly, low-energy density (ED) meals were shown to reduce energy intake during the present and succeeding meal [26], especially if the meal was high-protein (≥25–30 g), which particularly increased specific satiety (per kcal) and reduced energy intake [27].

Rapidly absorbed carbohydrates, mostly from refined grains/flours, have been shown to increase postprandial glycaemia [28,29,30], which in turn induced leukocyte activation [31] and advanced glycation end products (AGEs) [32], leading to endothelial damage and dysfunction [3] and ultimately enhanced inflammation [33]. Compared to lipid and protein, ingestion of glucose produced the largest increase in leukocyte gene expression of inflammatory mediators [34]; and high GI foods, characteristically refined/low-fibre carbohydrates, showed high inflammatory effects [35,36], and similarly with GL [37].

Adding protein to carbohydrates reduced glycaemic response as compared to a high-carbohydrate meal [38], and consuming vegetable and protein before carbohydrate reduced glycaemic and insulin levels, as compared with the reverse food order [39]. 

Acute food-induced oxidative stress, which is linked to inflammatory response, e.g., following high consumption of red meat, especially when combined with high polyunsaturated fatty acids (PUFA), including n-3 long-chain PUFA (LCPUFA) [40], was shown to be especially strong in older age and in people having metabolic dysfunction [41].

Recent studies have shown that ingestion of a meal high in saturated fat can induce metabolic endotoxemia [42,43] and increase the expression of proinflammatory immunoregulatory molecules such as lipopolysaccharides [42], TNFα, VCAM-1 [43] and IL-1β [44], as well as modulating circulating miRNAs controlling inflammatory and lipid and protein metabolism in the postprandial period [43], while increased unsaturated fatty acid consumption could attenuate the inflammatory status [44]. Intake of foods high in saturated fat was shown to postprandially influence blood lipid concentrations independently of body weight, and to more strongly influence glycaemic and proinflammatory lipopolysaccharides and metabolic risk [45].

A meal high in total fat that increased postprandial plasma lipemia was observed to further induce endothelial dysfunction, reinforcing the association with atherosclerosis [46], which independently predicts risk of cardiovascular events [47,48] more strongly than high morning/fasting triglycerides [47,49]. This has been attributed to generation of pro-inflammatory cytokines and resultant oxidative stress [1,50], while intervention with a low GI diet [51,52] and a low GL diet [53] reduced plasma CRP in short-term and long-term studies in overweight and obese adults [8].

The relationship of the three major post-prandial metabolic and immune responses—satiety, glycaemic and oxidative stress—to one another and ultimately to increased risks associated with inflammation, raised interest in the potential effect of meal composition and meal course sequence on optimizing postprandial outcomes. The traditional Mediterranean diet, and as applied to meals, was selected as a benchmark, as it has long been accepted as a ‘gold standard’ [54], shown to reduce obesity [55], non-communicable disorders such as diabetes mellitus [56], cardiovascular diseases [57,58], cancer [59] and dementia [60], had specifically demonstrated advantageous acute postprandial anti-inflammatory effects [61].

Illustrating the suggested meal order in three pyramids based on the three selected metabolic factors—satiety, glycaemic and oxidative responses—would be assumed to help in communicating the concept, demonstrate its applicability and encourage informed consumer adherence.

## 2. Materials and Methods 

Scientific literature describing the acute effects of foods, their components, interactions between them, and the meal course order on key metabolic axes—postprandial satiety, glycaemic and oxidative responses and associated immune outcomes—exemplifying their direct health effects and those potentially exerted through inflammatory response, was reviewed. Reference was made to the typical traditional Mediterranean meal components and course order, for their known positive metabolic and health effects. The conclusions of this review regarding potential effects of meal composition and sequence were illustrated in a representative ‘healthy meal sequence pyramid’ for each of the metabolic axes and for a composite of the three axes together. These pyramids offer educative visual communication of the suggested concept. 

## 3. Results

### 3.1. Satiety–Obesity Axis 

As satiety is a major factor in cessation of food consumption, in the obesogenic era the earlier it is attained, the better the control on food intake and resultant body weight, inflammatory response and health [62]. As satiety is first elicited by the gut distension response to portion size/volume/weight rather than to caloric/energy intake [63], the low energy density (ED, kcal/100 g) principle becomes a key factor in satiety-related intake control and obesity prevention [26] (Figure 1).

Non- and very low-caloric components, i.e., added/incorporated water and fibre, respectively, are highly and directly associated with satiety indices (SI, % of white bread per kcal) [64] and consuming pre-meal soups, fruits, vegetables and other low-ED preloads [64,65] has been shown to decrease overall caloric intake [66]. Even air incorporation enhanced satiety, supporting the gut distention effect [67].

Among caloric macronutrients, a satiating hierarchy—protein > carbohydrate > fat—shows that not all energy sources/macronutrients have the same impact on satiety. As protein induced a much stronger satiety level-per-kcal than fat and carbohydrate [68,69], allocating protein after foods/drinks high in added/incorporated water and fibre before high carbohydrate and/or fat foods—though fat delays gastric emptying—would reduce caloric intake in the present and succeeding meal [64]. 

#### 3.1.1. Non-Caloric Component

##### Added and Incorporated Water

Starting a meal with liquid/water courses and/or pre-loads high in added/incorporated water, e.g., in vegetables or vegetable salads [70], fruit [71], soup [22] and low-kcal beverages [72], was shown to effectively increase satiety and reduce immediate energy intake [73] in the present and ensuing meals [74] through contribution to gastric distension.

#### 3.1.2. Caloric Components 

##### Fibre

Fibre, the second-most satiating component per kcal after water, contributes greatly to gastric fullness-related satiety. It slows the rate of gastric emptying, and augments the release of neuroendocrine responses involved in satiety and food intake regulation [75,76]. 

Soluble fibres, particularly those with high viscosity, slow digestion and increased absorption of macronutrients, and extend the release of appetite-regulating hormones such as cholecystokinin (CCK) [77,78]. For example, β-glycan in oat raises satiety much more than highly processed ready-to-eat breakfast cereals such as Cornflakes^®^ [79]. In contrast, insoluble fibres, which provide bulk to the diet and increase the rate of transit through the small bowel, are less satiating and hunger-suppressing [80].

##### Protein

High-protein diets demonstrated a long-term impact on weight loss, mostly through its high satiating than iso-calorically dense high-carbohydrate or high-fat diets, even without changing net energy intake [21,81]. Gastrointestinal tract sensors register proteins as satiating [82] and stimulate gastrointestinal hormonal signalling [83] that reaches the brain [84], contributing to the protein sensory experience [85,86]. Amino acids in the gastrointestinal tract induce release of CCK [87,88] and anorexigenic substances such as glucagon-like peptide (GLP)-1 and peptide YY (PYY) [89,90,91]. Thus, prioritizing high-protein foods early in the meal sequence before carbohydrate and fat [68,83] would be expected to increase their satiating effect more than with the inverse order [92].

Plant Protein. Vegetarian meals (e.g., based on legumes) affect satiety sensations similarly or even favorably compared to animal-based meals (pork/veal) when having similar energy and protein content, even with lower fibre content. A ‘high protein (HP)-Legume’ meal was shown to induce greater fullness than ‘low protein (LP)-Legume’ and ‘HP-Meat’ meals [93], an advantage that is further augmented with higher fibre content [94,95]. Soy protein has demonstrated a favourable effect on satiety in obese animals and humans, reducing excess body and liver fat, insulin resistance, plasma lipids and associated risks [95]. 

Dairy Protein. Whole milk and yoghurt have been shown to increase circulating levels of the anorectic peptides GLP-1 and PYY. Yoghurt as a fermented milk with partially hydrolysed lactose, was more satiating than non-fermented milk-based beverages with no energy compensation at the next meal [96]. Whey protein, compared to casein, was associated with more rapid gastric emptying and a postprandial increase in plasma amino acid concentrations [97], whereas casein-derived peptides (casomorphins) reduced gastrointestinal motility and postprandial increase of plasma amino acids, which blunted their satiating effect [98,99,100]. Goat dairy showed a slightly higher satiating effect than cow dairy products [101]. Increased faecal fat and energy excretion following dairy calcium and protein intake may further explain why dairy products support weight loss [102].

Egg Protein. Egg has demonstrated very high satiating potential, likely associated with a GLP-1 peptide effect [103]. An egg-based breakfast has been shown to suppress ghrelin response [104], and to be associated with greater satiety and reduced food intake throughout the day compared to an isocaloric, equal-weight bagel-based or oatmeal breakfast [105]. This supports the advantage of allocating egg protein earlier in the meal to reduce ensuing and total energy intake [106].

Fish and Seafood Protein. Fish has long been considered to have great satiating capacity [107], especially low-fat fish, which has demonstrated a very high SI value [64] and potential for reducing energy intake at the subsequent meal [108]; while fatty fish consumption was associated with increased body weight and waist circumference [109]. Fish satiating capacity is assumed to be related to serotonergic activity induced by high postprandial ratio of tryptophan to neutral amino acids [110], and high n-3 LCPUFA in fish and seafood were shown to effectively reduce hunger and increase fullness sensations [111].

Meat Protein. No differences for hunger, satiety, fullness or appetite score were found with isocaloric, fibre-matched meals based on fava beans/split peas compared to veal/pork/eggs [112]; though the latter yielded lesser satiating sensations, compared to a fibre-rich vegetarian-based meal [93]. 

##### Carbohydrate

Though whole/complex carbohydrate foods often yield lower risk of overweight/obesity compared to high fat consumption [113,114], a high-carbohydrate diet showed only transiently higher satiety response than high-protein and high-fat diets [21]. 

Satiety was inversely associated with GI [115,116] and hedonic/palatability (i.e., with sugar) [64], expressed as ‘wanting’ and ‘liking’ [117] beyond fullness-related satiety. 

Thus, low-glycaemic/slow-release carbohydrate foods with high-fibre and high-protein foods—e.g., whole grains mixed with beans—would be the recommended type of carbohydrates for increasing satiety and reduced caloric intake and GL. 

As the pleasure of sweet taste is innate and universal across individuals, ages, races and cultures, and spontaneous intake is increased when freely available [118], refined sugars/sweets would preferably be served last in the meal, and consumed—if at all—in minimal amounts.

##### Fat

High-fat foods tend to drive greater unconscious energy intake than high-carbohydrate foods [119], a phenomenon termed ‘high-fat hyperphagia’ or ‘passive overconsumption’ [120], without increased sensations of satiety [69]. Its inverse association with satiety [64] is related to its high ED and low fullness potential per kcal [65], and contribution to palatability and hedonic response [121,122]. For example, high-fat yoghurts demonstrated a lower appetite-suppressive effect than high-carbohydrate yoghurts [123]. High-fat diets further alter gut microbiota potentially associated with weight gain and increased body fat [124]. Thus, reducing and delaying fat toward the end of the meal would be advised for circumventing its satiety reducing potential. 

### 3.2. Glycaemic Axis 

Postprandial hyperglycaemia increased oxidative stress, hypercoagulability, endothelial dysfunction, and inflammation [125]. Chronic hyperglycaemia, as reflected by glycated haemoglobin (HbA1c), is considered to be a major risk factor for chronic inflammation and sequelae [126]. Glycaemic variability, or high postprandial glucose peak, was recently suggested to be even more important than HbA1c [127] or single-time point measurements, e.g., morning fasting blood glucose, for predicting cardiovascular risks [128]. Thus, reduced postprandial hyperglycaemia is an important preventive and therapeutic goal against vascular diseases, type 2 diabetes mellitus, and additional inflammation-related non-communicable diseases [125,128] (Figure 2).

Glycaemia or the glycaemic response [129] can be greatly reduced by adding protein, fat and vegetables [38,130,131]. Some culinary plants reduce glucose response via delayed gastric emptying rate, e.g., cinnamon [132], and/or due to their antioxidant content [133], e.g., polyphenols from red grapes [134].

#### 3.2.1. Macronutrients

##### Carbohydrate

Both quantity and type of carbohydrate affect blood glucose levels [116,128]. Low-carbohydrate meals/diet reduce blood sugar responses [135,136,137]. In type 2 diabetes mellitus, low-carbohydrate diet demonstrated an advantage over high-carbohydrate, high-protein, vegetarian/vegan, low-GI, high-fibre, and Mediterranean type diets [138]. Even extremely low carbohydrate intake, as in the Spanish ketogenic Mediterranean diet study, was suggested to be a safe and effective way to improve fasting blood glucose levels [139], meliorate related health risks. A study employing a combination biphasic ketogenic Mediterranean diet and Mediterranean diet maintenance protocol yielded similar results, and was shown to improve long-term compliance [140]. 

Beyond sugar/sucrose (glucose+fructose) having a lower GI and insulin response (IR) than glucose alone, due to the low GI of fructose [129], increasing intake was associated with decreasing insulin responsiveness, in contrast to progressive plasma insulin responses to all carbohydrates [141]. Moreover, food fructose consumed in very high amounts can contribute to increased triglycerides [142], aglycation and inflammation [143], which are highly deleterious.

Reduced glycaemic response when carbohydrate intake followed that of vegetables and proteins was described as “comparable to pharmacological agents” in adults [144] suggesting food intake order as a sound strategy for glycaemic control [39]. 

##### Fibre

Dietary fibre, especially soluble, lowers blood sugar responses [145,146], and resistant starch has been shown to improve glucose metabolism and attenuate related inflammation in individuals with type 2 diabetes [147]. Meals based on fibre-rich rye sourdough bread, salmon, vinegar and vegetables, were associated with approximately 40% lower blood glucose and insulin responses, compared with a fast food/meal matched for energy and macronutrients [148]. A cross-over study in adults at risk for the metabolic/dysglycaemic syndrome showed whole grains to provide anti-inflammatory protection compared to refined grains, an advantage attributed in large part to fibre [149].

##### Protein

Protein increases insulin secretion, resulting in reduced glycaemic response [150]. This was shown by adding chicken meat and salad to mashed potatoes that significantly reduced the meal’s GI or GL (GI multiplied by the amount of carbohydrate) [151]; adding beans to rice has a similar effect [152,153]; protein ‘pre-load’ has acutely improved glucose tolerance, mainly by delaying gastric emptying and enhancing insulin secretion [154,155,156,157]. Pre-meal whey protein reduces post-meal glucose levels, possibly by slowing gastric emptying and increasing release of gut peptides, including incretins, which support glycaemic control [155,158,159]. Whey amino acids, like pre-digested proteins in yoghurt, directly stimulate β-cells to secrete insulin, which enhance satiety and glycaemic control [158]. Effect of lean red meat was similar to low-fat dairy protein [160]. However, a conventional beef hamburger meal showed significantly higher glycaemic and insulinemic response than a salmon burger meal despite both having similar glycaemic load [148]. Together, the evidence suggests that both plant [152,153] and animal proteins have the potential to reduce glycaemic response, especially when consumed in an early stage of the meal [161,162,163].

##### Fat

Fat’s capacity to delay gastric emptying and augment insulin secretion explains why a fat preload can acutely improve glucose tolerance [154,157]. As fats usually take the longest to digest, adding olive oil alone to a low-glycaemic diet was shown to further reduce fasting blood glucose and resulting aglycation, as indicated by lower HbA1c [164].

#### 3.2.2. Vegetables, Spices and Phytonutrients

High total daily vegetable intake, particularly of green vegetables, has been correlated with improved control of triglyceride and HbA1c levels in elderly type 2 diabetes patients [165]. Eating vegetables before carbohydrates has repeatedly been shown to reduce glucose and insulin response [144,166], blood glucose fluctuations and the insulin cascade—as compared to the reverse order. The effect was shown by acute (postprandial) [144] and long-term (through 2.5 years) follow-up [166]. Even plant polyphenols alone could reduce glycaemic and insulinemic responses in diabetic patients [167] and improve metabolic aging in obese and inflammatory conditions [168].

#### 3.2.3. Fermentation and Organic Acids

Vinegar (acetic acid), a common ingredient in traditional Mediterranean first courses, slows down the carbohydrate digestive process by inhibition of gastric emptying, thus preventing blood sugar spikes [169]. Recent studies showed that adding 1 to 2 tablespoons of vinegar to a meal containing white bread or white rice can lower postprandial blood glucose by 25–35% and increase post-meal satiety by more than 2-fold [170]. Additionally, lactic acid contributes to reduce GI, e.g., as shown by bread containing lactic acid produced during the sourdough fermentation or added directly to the bread dough that lowered the postprandial glucose and insulin responses [171]. Addition of fermented milk (yoghurt) and pickled cucumber to a high GI breakfast significantly lowered postprandial glycaemia, compared to adding regular milk and fresh cucumber [172]. Moreover, combining lactic acid bacteria with polyphenols has demonstrated significant improvement of inflammation induced by a high-fat diet in a murine model through their synergistic effect on glucose and insulin metabolism [173].

### 3.3. Oxidative Stress Axis 

Oxidative stress and inflammation are closely interrelated, tightly linked to pathophysiological processes [174,175], accentuating each other and inducing progressive damage and disease [176] (Figure 3).

As the oxidative process is a chain reaction, early dilution of gastric content and introduction of antioxidants throughout the meal would protect the stomach and body against oxidative stress throughout the meal, along the digestive system and following absorption of oxidized lipids remnants. This is especially important before foods reach the stomach, where the high-acid environment facilitates food oxidation [177]. Such protection can also be achieved when these measures are supplied together with pro-oxidative factors in the meal [178], as recently described by the postprandial oxidative stress (POS) index for a portion of Greek salad or red wine [40].

#### 3.3.1. Macronutrients

##### Protein

Animal-derived foods that are rich in fat and protein are generally high in advanced glycation end-products (AGEs) produced during cooking, unless cooked in moist heat or water in a covered pot at low temperatures and for a short time [179]. High temperatures and extended cooking, e.g., well-done vs. medium-rare preparation [180], increase the risk of oxidative stress. Red meat induces lipid peroxidation in the stomach, resulting in POS and reactive cytotoxic aldehydes such as malondialdehyde (MDA)—indicators of oxidative stress—which are absorbed in the blood and react with cell proteins, DNA and lipids, and drive inflammation and related dysfunctional responses, strongly pronounced when combined with oxidative-prone PUFA, e.g., n-3 and/or n-6 PUFA [40]. This explains why low red meat intake, which is a major characteristic of the Mediterranean diet, eliminates a major source of pro-oxidants and pro-inflammatory trajectory, and how drinking red wine and consuming vegetables together with red meat provide a protective effect [181].

##### Starchy Carbohydrate

High blood glucose potentially generates radical oxygen species (ROS) through mitochondrial oxidation, nicotinamide adenine dinucleotide phosphate (NADPH) oxidase, sorbitol and activated glycation [182]; increased polymorphonuclear leucocytes and mononuclear cells, as shown following vitamin E oxidation [183]; and increased inflammation through ROS enhancing expression of TNF-α, IL-6 and plasminogen activator inhibitor (PAI)-1 [184].

High dietary GI and GL associations with oxidative stress were shown by urinary F2-IsoP and 15-F2t-IsoP-M concentrations [185]. Isolated fructose from refined sugar raises uric acid [186], which generates ROS by interacting with peroxynitrite [187] and oxidized lipids [188]. Additionally, high insulin response increases membrane enzymes supporting generation of hydrogen peroxides [189]. A clinical study demonstrated a significant association between high GI and corresponding Dietary Inflammatory Index (DII^®^) [8].

##### Fats and Fatty Acids

Saturated and trans Fatty Acids (SFA, t-FA). SFA and tFA stimulate a proinflammatory state contributing to insulin resistance. These fatty acids can be directly involved in several inflammatory pathways, autoimmunity, allergy, cancer, atherosclerosis, hypertension and heart hypertrophy, as well as other metabolic and degenerative diseases; and indirectly through alteration in the gut microbiota, which may be associated with endotoxemia and inflammation [190,191].

Monounsaturated Fatty Acids (MUFA). Extra-virgin olive oil (EVOO) high in the n-9 MUFA oleic acid (18:1) was associated with reduced risk of cardiovascular disease and total mortality [192], postprandial glucose, low-density lipoprotein (LDL) cholesterol oxidation, inflammation, endothelial adhesion molecules, and total oxidative stress in healthy and hypertriglyceridemia subjects [193,194,195]. In the presence of red meat induced oxidation, olive oil protects vitamin E, while fish oil increased vitamin E oxidation [196]. Beyond the oxidative and inflammatory protection of high of MUFA:SFA and low n-6:n-3 PUFA ratios [197,198], the antioxidative effect of olive oil is also attributed to the high content of polyphenols, e.g., hydroxytyrosol and oleuropein [199].

Polyunsaturated Fatty Acids (PUFA, n-3 and n-6). PUFAs that are absorbed in the postprandial state are incorporated into lipoproteins, e.g., LDL, where they may promote LDL oxidation, generate lipid hydroperoxides, contribute to pro-oxidative stress and endothelial dysfunction, and increase metabolic and inflammatory risk [178].

Though both high n-6 and n-3 are highly oxidizable, especially in the acidic gastric environment, n-6 linoleic acid (LA) is enzymatically transformed into arachidonic acid and resultant n-6 pro-inflammatory eicosanoids, while n-3 PUFAs/LCPUFAs from plant and/or marine sources competitively reduce the n-6 inflammatory trajectory [200]. Correspondingly, a diet low in n-6 PUFA-rich oils (e.g., corn, sunflower and safflower), and high in n-3 PUFA plant sources (e.g., green leafy vegetables, chia, and flax/linseed) and/or n-3 LCPUFA from animal sources (e.g., eicosapentaenoic acid [EPA] and docosahexaenoic acid [DHA]) from fatty fish and n-3 fortified eggs, can together reduce responsive cellular and plasma concentrations of inflammatory factors [201].

#### 3.3.2. Food Antioxidants

Carotenoids (e.g., β-carotene and lycopene), as from carrot, pumpkin, tomatoes, and red pepper, have demonstrated protection against oxidation, free radicals, and inflammatory factors [202]. Vitamin C attenuates postprandial lipæmia-induced oxidative stress and endothelial dysfunction, e.g., after a fatty meal, especially in people with diabetes [203]. Vitamin E, selenium, glutathione, and phytochemicals such as capsaicin from red pepper, resveratrol from red wine, EGCG in green tea, hydroxytyrosol and oleuropein from virgin olive oil, stilbenes and lignans from whole grains and legumes, and catechins, flavonol glycosides, anthocyanins and procyanidins in cocoa [197], all add antioxidative and anti-inflammatory capacity.

##### Spices and Seasonings

Plant-based seasonings are important sources of dietary antioxidants, and their addition to food has been shown to attenuate oxidative risk and inflammation. Adding a spice mixture to hamburger meat prior to cooking reduced urinary MDA, increase urinary nitrate: nitrite ratio, and improve postprandial endothelial function [204]. Turmeric combined with black pepper significantly decreased lipid peroxidation in hamburger meat [205], red wine polyphenols reduced the absorption of MDA from cooked turkey meat [206], and tomato powder contributed an antioxidative effect to cooked pork [207]. Cumin and fresh ginger [208]—commonly used in meat dishes—together provide protective antioxidant to meat products [209].

Herbs, spices, and seasonings are major components of Mediterranean cuisine, and believed to contribute greatly to the antioxidative and anti-inflammatory properties of this dietary pattern [210]. Typical spices in Mediterranean meat dishes include cinnamon, garlic, and rosemary, which have been shown to inhibit AGE formation in vitro and in animal models [211]. Lemon balm and marjoram increased the antioxidative capacity of salad [212], used as a traditional side dish in meat-inclusive meals. Red wine, a familiar feature of the Mediterranean diet [213,214] was associated with a 75% reduction in plasma MDA levels, if consumed during a red meat meal [215].

Similarly, antioxidants in whole grains [216] contribute to prevention of oxidative stress and active free radical formation as compared to refined/processed grain products [202]. Consuming food antioxidants earlier in the meal will be most effective in reducing the oxidative chain reaction.

Altogether, abundance of antioxidants is suggested to be a central beneficial factor of the Mediterranean diet [217,218], contributing anti-inflammatory effects as shown by reduced CRP and IL-6) [210], when compared with typical North American and Northern European dietary patterns [218].

#### 3.3.3. Food Preparation and Processing

Principal cooking recommendations to lower meat-derived pro-oxidants—heterocyclic amines, and AGEs—include short time, low temperature and covered heating, and adding plant foods with their skins and seeds, which are particularly rich in antioxidants and anti-inflammatory factors [202]. Herbs and spices, especially in marinades, have demonstrated effective reduction of free radical formation in meat cooking [219,220].

## 4. Discussion

Increasing understanding of the role of the postprandial states on health has yielded a significant depth of research, including into the impact of dietary obesogenic risks on inflammation-related responses. This paper appears to be the first to combine three major metabolic axes affected by meal consumption order—satiety-related food intake and resultant obesity, glycaemia and oxidative stress—which contribute to systemic inflammation, DNA mutations, metabolic errors and mental ageing, known to be leading risk factors in the aetiology of the modern chronic diseases and their epidemics [221].

A focus on the acute meal effect is important, due to the fact that in the modern world, people are increasingly found to be in a ‘postprandial’ state, resulting from increasing exposure to eating opportunities throughout the day, propagating the effects of western meals on cumulative metabolic trends [222]. Addressing these key related factors has already been shown to favourably affect postprandial outcomes [23,126,215,223,224].

The illustrative pyramids—here used to demonstrate the representative meal sequence according to each axis (Figure 1, Figure 2 and Figure 3)—provide a visual demonstration that would help apply and attain improved health outcomes beyond conventional dietary guidelines and recommendations.

Though the three pyramids rank all types of food, the basal/bottom levels are highly recommended to be consumed as introductory for every meal, while the upper/top levels—which include ultra-processed and high glycaemic and oxidative foods—would be better reduced, and/or only occasionally consumed, sparingly (in minimal portions), if at all.

Notably, some aspects of the three axes are quite consistent with one another, as demonstrated by the composite pyramid (Figure 4). For example, vegetables being first, due to their combined contribution of high satiating potential [64], antioxidative capacity [223] and reduced glycaemic impact [224]. Further food qualities, including whole food structure and selected components e.g., fibre and phytonutrients, and recommended preparation methods such as light cooking, together with a variety of plants, herbs, olive oil and spices, all positively influence the three axes.

Great emphasis is placed on the non- and very low-caloric macronutrients—water and fibre, respectively—due to their direct effect on gastric distention and fullness feeling that stimulate the satiety response in the brain [225]. Water and fibre further dilute and temporarily reduce gastric acidity, which otherwise would immediately induce oxidative trajectory [177], thereby reducing high inflammatory and glycaemic-related oxidative stress [226]. Fibres similarly affect satiety, hold gastric water, provide antioxidants, and further support a healthy gut microbiome [76,78,147,216,227], which may potentially regulate obesity, glycaemic and inflammatory responses [228].

Protein foods—from both plant and animal sources—are recommended to be the first macronutrient consumed in the meal, before carbohydrates and fats, to support earlier satiating potential [81] and reduce glycaemic response [151]. Light proteins with high relative satiating potential, including from plants (especially legumes) [229] and low-fat dairy (especially whey) have been associated with reduced hunger and glycaemic response [230]. Fish and white meats have been shown to be advantageous among animal flesh foods, e.g., against oxidation [231]. Similarly, light cooking of red meats—i.e., medium-rare rather than well-done—and slowly cooking with a cover to reduce temperature and exposure to air oxygen, cooked together with vegetables and their plant-based antioxidants—would be expected to attenuate related metabolic risks [232].

The advantage of delaying high-carbohydrate foods to a later stage in the meal is due to their medium-to-low SI [64], high glycaemic and insulinemic effects [129,144], and indirect potential to increase oxidative stress [127,233]. Reducing and delaying fatty foods to a late stage in the meal is supported by their unwanted postprandial lipemic effect [128], inverse association with satiety due to their low ED and contribution to palatability [64] and delayed gastric emptying [154]. However, advancing some fat/oil in the meal may reduce glycaemic response, and some oils, like olive oil, would also reduce oxidative stress, due to its stable n-9 MUFA oleic acid and high content of antioxidants [199].

Optimal fat composition of a meal for protection against oxidation would imply greater intake of MUFA (e.g., from olive, avocado, almonds and rapeseed) and protected n-3 LCPUFA—from both marine (e.g., fish and seafood) and land-based (e.g., n-3 fortified eggs, green leafy vegetables such as purslane [234], walnuts, flax and chia seeds [235]) sources, and controlled/restricted amounts of n-6 PUFA (e.g., from corn, sunflower, soy, and safflower oils), which are known for their pro-inflammatory and pro-coagulation effects [236].

The three axes of the ‘healthy meal order’ similarly emphasize the deleterious effect of processed foods and beverages – high in refined sugars that reduce satiety [237], are pro-oxidative [179] and enhance glycaemic response [64] with resultant production of pro-inflammatory factors, e.g., AGEs [33]—together support reducing and delaying confectionary foods and sweets to the very end of the meal.

As the selected axes were similarly applied and have demonstrated advantages in the traditional Mediterranean diet, which leads in association with health and longevity, it can be acknowledged that the Mediterranean meal course order makes a significant contribution to its success and becoming the ‘gold standard’ diet. Specifically, the traditional Mediterranean diet, having developed in a hot and sunny climate, habitually comprises starters high in water for rehydration at the beginning of the meal, e.g., soup (hot or cold) and vegetables, and a menu overall high in innate and added/incorporated water, including vegetables, fruits and liquid dairy, which support the three axes. Here ‘home cooking’ with water further explains the advantage over exposure to ultra-processed/condensed/dried, convenience and fast foods.

This traditional meal composition and sequence may have contributed to the link between adherence to the Mediterranean-type diet and improved cardiometabolic health [238,239], reduced risks of chronic diseases such as diabetes, cancer and dementia [218], and to the inverse association between an ‘olive oil and salad type diet’ and mortality, as compared to direct association with a ‘pasta and meat’ pattern [130].

The ‘composite pyramid’ (Figure 4) combines the goals of high-satiety, low-glycaemic, and low-oxidative responses for comprehensive anti-inflammatory guidance. It adds food intake orientation (daily, weekly, monthly, sparingly and/or occasionally) for optimal outcomes. The representative meal course order can be used as an independent guide and/or extension of the various types of food guides, e.g., the Mediterranean diet pyramid based on the original United States Department of Agriculture (USDA) model [54,240], as well as guides from representative Mediterranean countries (Greece, Spain and Italy), that aimed to provide the populations with dietary advice for prevention of chronic diseases and to promote healthy longevity [240]. Designing meals according to the suggested axes could encourage better adherence to healthy dietary recommendations, thereby preserving this Mediterranean ‘health heritage’ and old nutrition wisdom that is promoted by UNESCO [241], highly relevant and applicable today.

## 5. Conclusions

The main contribution of the present paper is the emphasis put on acute effects of meals expressed by the postprandial state, and the newly developed metabolic concept of meal sequencing according to three key factors, satiety, glycaemia, and oxidative stress, corresponding to comprehensive potential for improvement in postprandial state and resulting reduced chronic inflammation and related health risks.

The high similarity among recommendations in the three axes—first courses high in non- and very low-caloric components high in added/incorporated water and with fibre, respectively (e.g., soups and low-ED preloads) for initiating gastric fullness-related satiety, followed by vegetable-based/rich dishes; then lightly-cooked protein foods to enhance satiety, and reduce glycaemic and oxidative responses—support the potential for attaining a combined health effect as reflected in an healthy anti-inflammatory composite pyramid integrating the three-axes sequencing principle. Comprehensive research evidence supporting the metabolic concept of meal sequence provides a variety of options to attain the healthy meal goals. The compatibility of the presented sequences with the traditional Mediterranean meal suggests their inherent contribution to overall success of this diet. Using ‘food pyramids,’ which are highly familiar tools in nutrition education and guidelines, would be expected to help with adopting the principle and to increasing adherence to the suggested concept and meal sequence, warranting evaluation of application and outcomes.

## Figures and Tables

**Figure 1 nutrients-11-02373-f001:**
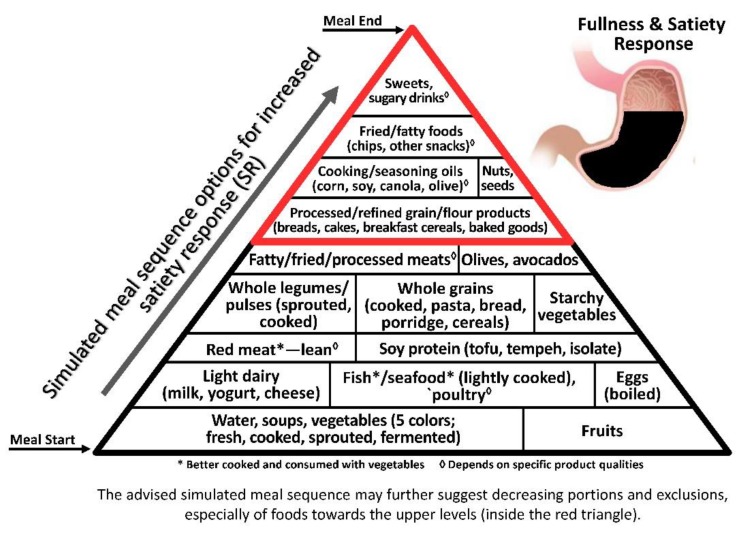
The Meal Sequence Pyramid per Satiety Response (SR) Axis. Foods are ranked according to fullness-related satiety per contribution to gastric filling (fullness proportional to stomach cavity), protein, fibre and hedonic aspects. Foods at the bottom of the pyramid that contribute to the greatest SR, being highest in innate/added/incorporated water and fibre and therefore *w*/*v* per kcal, are recommended to start the meal and prevent overconsumption; the next levels include foods high in innate/added/incorporated water and protein, which affect both biochemically induced satiety-related hormones as well as *w*/*v* and therefore scored medium SR; and foods at the highest levels are the most energy-dense, highly processed and palatable foods, which contribute the least to fullness sensation per kcal and therefore would best be limited and scheduled toward the end of a satiating meal, when hunger is least likely to add to the motivation for further consumption.

**Figure 2 nutrients-11-02373-f002:**
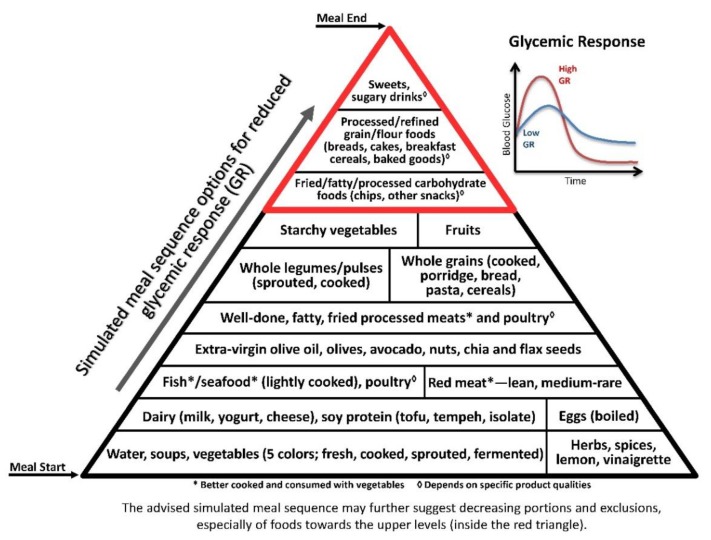
The Meal Sequence Pyramid per Glycaemic Response (GR) Axis. This pyramid ranks foods according to their potential to cause increase in blood glucose postprandially, similarly to the GI [129]. Foods at the bottom of the pyramid contribute to the lowest GR, being lowest in carbohydrate concentration (specifically, soups, low glycaemic preloads and vegetables); the next levels include foods high in innate/incorporated water, fibre and protein (e.g., legumes, dairy and fish; whole cooked grains/cereals and other complex carbohydrate foods), as well as fat, that together contribute to reduced GR; and the highest levels include foods high in GI, processed carbohydrates and/or sugars and low in protein and/or fat, therefore inducing the sharpest GR―being undesirable and/or that should be delayed to the end of the meal, after attenuation by vegetables and foods high in protein and fat.

**Figure 3 nutrients-11-02373-f003:**
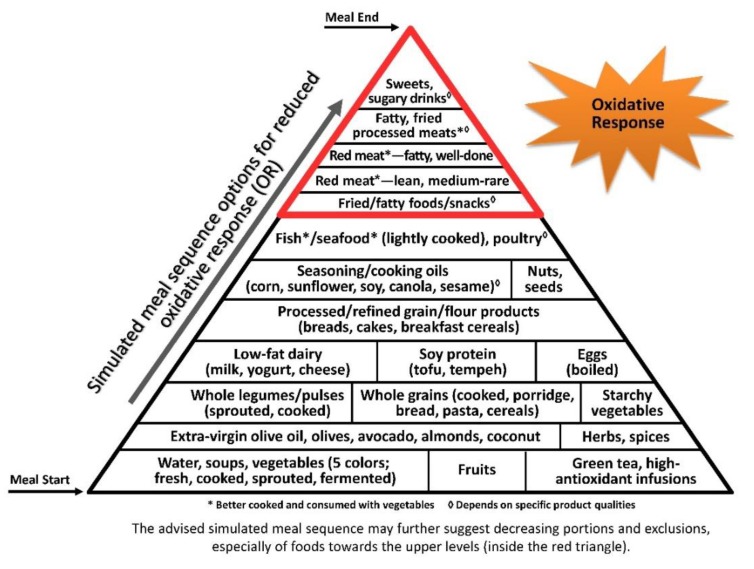
The Meal Sequence Pyramid per Oxidative Response (OR) Axis. Foods are ranked according to their potential to enhance reactive oxidation species (ROS) postprandially. Foods at the bottom of the pyramid have the lowest OR, being the least oxidizable and highest in antioxidants that neutralize ROS, and are recommended for starting the meal and preparing early protection against OR from ensuing courses; the next levels include relatively whole, minimally processed foods low in oxidative compounds, further contributing to a moderate OR; and the highest levels include highly processed foods through technologies that generate high amounts of ROS, with little-to-no antioxidant content, and foods high in pro-oxidative potential like fried foods and high-iron red meat, therefore contributing to the sharpest OR, and should be delayed to the end of the meal (if consumed at all), after the body has been primed with attenuating antioxidative cascades from earlier courses.

**Figure 4 nutrients-11-02373-f004:**
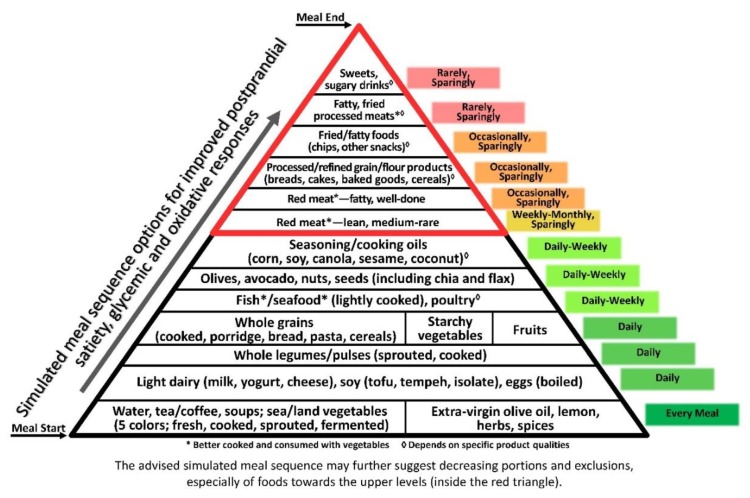
The composite Healthy Meal Sequence Pyramid per Satiety, Glycaemic, and Oxidative Responses. It provides a composite ranking of foods in the meal and recommended frequency of incorporation into diet and lifestyle, per their combined effects on satiety, glycaemic, and oxidative responses. Foods at the bottom of the pyramid are those minimizing the three response axes, and are recommended to start each meal, to protect the body from cumulative deleterious responses from ensuing courses. Foods at the next several levels generate minimal-to-moderate responses and continue to contribute protection; and foods at the top levels of this pyramid are recommended to be either avoided or used only sparingly/occasionally, and preferably at the end of the meal, when the body has been primed against hunger-driven overconsumption and sharp GR and OR.

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
