# Peer review of "The Metabolic Concept of Meal Sequence vs. Satiety: Glycemic and Oxidative Responses with Reference to Inflammation Risk, Protective Principles and Mediterranean Diet"

_nutrients, 2019, doi:10.3390/nu11102373_

Round 1
Reviewer 1 Report
Shapira, in your manuscript entitled “The metabolic concept of meal sequence vs. satiety, glycemic and oxidative responses, with reference to inflammation risk, protective principles and Mediterranean diet” had the objective to review regarding potential effects of meal composition and illustrating in a ‘healthy meal sequence 100 pyramid’ for each of the metabolic axes and for a composite of the three axes together. However, the scope of the study is good, recent studies about postprandial state and dietary fatty acid have focused in metabolic and immune axes. These studies and references have not been included and reviewed. The novelty and significance of content is poor.
Author Response
Thank you very, very much. The requested studies have been referenced, and indeed greatly enhance the background of the paper.
The following is the added text:
Recent studies have shown that ingestion of a meal high in saturated fat can induce metabolic endotoxemia [42,43] and increase the expression of proinflammatory immunoregulatory molecules such as lipopolysaccharides [42], TNFα, VCAM-1 [43] and IL-1β [44], as well as modulating circulating miRNAs controlling inflammatory and lipid metabolism proteins in the postprandial period [43], while increased unsaturated fatty acid consumption could attenuate the inflammatory status [44]. Intake of foods high in saturated fat was shown to postprandially influence blood lipid concentrations independently of body weight, and to influence glycemic in obesity; when basal blood triglyceride concentration was high in obese individuals, the increase was greater, promoting increased proinflammatory lipopolysaccharides and metabolic risk [45].
Added references:
Lopez-Moreno, J.; Garcia-Carpintero, S.; Jimenez-Lucena, R.; Haro, C.; Rangel-Zuniga, O.A.; Blanco-Rojo, R.; Yubero-Serrano, E.M.; Tinahones, F.J.; Delgado-Lista, J.; Perez-Martinez, P., et al. Effect of Dietary Lipids on Endotoxemia Influences Postprandial Inflammatory Response. J Agric Food Chem 2017, 65, 7756-7763, doi:10.1021/acs.jafc.7b01909. Quintanilha, B.J.; Pinto Ferreira, L.R.; Ferreira, F.M.; Neto, E.C.; Sampaio, G.R.; Rogero, M.M. Circulating plasma microRNAs dysregulation and metabolic endotoxemia induced by a high-fat high-saturated diet. Clin Nutr 2019, 10.1016/j.clnu.2019.02.042, doi:10.1016/j.clnu.2019.02.042. Monfort-Pires, M.; Crisma, A.R.; Bordin, S.; Ferreira, S.R.G. Greater expression of postprandial inflammatory genes in humans after intervention with saturated when compared to unsaturated fatty acids. Eur J Nutr 2018, 57, 2887-2895, doi:10.1007/s00394-017-1559-z. Alayon, A.N.; Rivadeneira, A.P.; Herrera, C.; Guzman, H.; Arellano, D.; Echeverri, I. Metabolic and inflammatory postprandial effect of a highly saturated fat meal and its relationship to abdominal obesity. Biomedica 2018, 38, 93-100, doi:10.7705/biomedica.v38i0.3911.
Reviewer 2 Report
Review of Nutrients #568451
This is a scholarly narrative review summarising recent findings on foods and food factors that reduce glycaemia and oxidative stress, while increasing satiety. The author rightly focuses on acute postprandial effects on the basis that humans are mostly in a postprandial state over 18 hours a day. The conclusion is that meal order is critical so that foods that increase satiety are consumed first, followed by those that reduce glycaemia and oxidative stress. Admirably, the author is careful to qualify ‘statements of fact’ with a reference, and thus there 248 publications in the reference list.
Specific comments for authors
Line 7. Please include the city and country. Line 11. The author must state that this is a narrative review (rather than a systematic review). The sentence could be re-phrased as: In this narrative review, meal components…..were explored in relation to postprandial responses….etc Line 53. Please state increased food variety and diet quality IS ASSOCIATED WITH lower obesity... Line 63. Please delete AND SUGARS. In general sugars (and sugary foods) are digested and absorbed at a slower rate than most starchy foods (1) Only glucose is absorbed rapidly but it is rarely used as a food ingredient. Sucrose is cheaper and has a GI of 60-65. Figure 1 /Line 113. It is rather incongruous/illogical to suggest that water alone is satiating. Do you mean high moisture foods such as fruit and vegetables? I think the diagram and the others would have more scientific credibility if you removed water. Line 120. See point 5 above. Fibre yields calories, via fermentation in the large bowel and absorption of the short chain fatty acids. For food labelling purposes, most countries assume fibre has ~2 calories per gram. Line 165. To my knowledge yogurt does not contain pre-digested protein (down to amino acids). Some of the lactose (not all) has been hydrolysed. Line 201. I think it’s a good suggestion to serve sweets last but to suggest they should not be consumed at all sets people up to fail and feel guilty. As you say, sweetness is innate. In the past, honey was widely consumed on a daily basis (2). Line 225. Responsive hyperglycaemia??? Do you mean chronic hyperglycaemia? Line 240. See point 4 above. It is not correct to say that GI can be reduced by adding protein etc. The wording should be ‘Glycaemia or the glycemic response can be reduced’. Line 257. Only when fructose is consumed in very large amounts (higher than the 95th percentile in US) does it contribute to higher TG (3). Line 271. This sentence doesn’t make sense. Do you mean protein increases insulin secretion, resulting in reduced glycaemic response? Line 285. AUGMENT insulin secretion would be preferable to ENHANCE. Line 299. The major mechanism by which vinegar reduces glycaemia is by strong inhibition of gastric emptying, not inactivation of enzymes (4). Line 387. It is preferable to say PEOPLE WITH DIABETES rather than DIABETICS. LINE 423. It’s best not to claim first time unless you have undertaken a systematic review. Line 432. If something is important, you don’t need to qualify it with HIGHLY. LINE 442. It is unrealistic and wasteful to suggest that processed food is not consumed at all. All food is processed these days, even fresh fruit and vegetables which are stored under modified atmospheres to extend shelf-life and reduce wastage. Even hunter-gatherers processed and stored foods. Figures – these don’t need a box around them.
References
Atkinson F, Foster-Powell K, Brand-Miller J. International tables of glycemic index and glycemic load values, 2008. Diabetes Care 2008;31:2281-3 Marlowe FW, Berbesque JC, Wood B, Crittenden A, Porter C, Mabulla A. Honey, Hadza, hunter-gatherers, and human evolution. Journal of Human Evolution 2014;71(0):119-28. doi: http://dx.doi.org/10.1016/j.jhevol.2014.03.006. Ludwig DS, Hu FB, Tappy L, Brand-Miller J. Dietary carbohydrates: role of quality and quantity in chronic disease. BMJOpen 2018;361. doi: 10.1136/bmj.k2340. Johnston CS, Steplewska I, Long CA, Harris LN, Ryals RH. Examination of the Antiglycemic Properties of Vinegar in Healthy Adults. Annals of Nutrition and Metabolism 2010;56(1):74-9.
Author Response
Thank you very, very much. The requested changes have been made as follows:
Please include the city and country: added
The author must state that this is a narrative review (rather than a systematic review). The sentence could be re-phrased as: In this narrative review, meal components…..were explored in relation to postprandial responses….etc: Statement added as specified
Please state increased food variety and diet quality IS ASSOCIATED WITH lower obesity...Statement added as specified
Please delete AND SUGARS. Deleted
It is rather incongruous/illogical to suggest that water alone is satiating. Do you mean high moisture foods such as fruit and vegetables? The term was corrected to "incorporated water," to reflect the concept of moisture while referring to water as a macronutrient.
Fibre yields calories, via fermentation in the large bowel and absorption of the short chain fatty acids. Accordingly "non-caloric "was modified to "very low-caloric"
To my knowledge yogurt does not contain pre-digested protein (down to amino acids). Some of the lactose (not all) has been hydrolysed. "Pre-digested protein" has been replaced by "partially-hydrolysed lactose"
I think it’s a good suggestion to serve sweets last but to suggest they should not be consumed at all sets people up to fail and feel guilty. Agreed, and elimination suggestion removed
Responsive hyperglycaemia??? Do you mean chronic hyperglycaemia? Yes, modified
It is not correct to say that GI can be reduced by adding protein etc. The wording should be ‘Glycaemia or the glycemic response can be reduced’. Wording modified as recommended
Only when fructose is consumed in very large amounts (higher than the 95th percentile in US) does it contribute to higher TG. Specification added
This sentence doesn’t make sense. Do you mean protein increases insulin secretion, resulting in reduced glycaemic response? Yes, modified
AUGMENT insulin secretion would be preferable to ENHANCE. Replaced as recommended
The major mechanism by which vinegar reduces glycaemia is by strong inhibition of gastric emptying, not inactivation of enzymes. Corrected as noted
It is preferable to say PEOPLE WITH DIABETES rather than DIABETICS. Corrected as noted
It’s best not to claim first time unless you have undertaken a systematic review. Statement modified to be more specific: "This paper appears to be the first to combine three major metabolic axes affected by meal consumption order..."
If something is important, you don’t need to qualify it with HIGHLY. "Highly" removed
It is unrealistic and wasteful to suggest that processed food is not consumed at all. "Ultra"processed specified
Figures – these don’t need a box around them. Boxes removed
Reviewer 3 Report
This was a very well-written paper that provided significant information. Very little information exists regarding diet-induced inflammation as a result of different food choices. This paper provided significant gap to the research regarding the types of foods that should be consumed pre and post-meals.
Author Response
Thank you very, very much.